# Machine Learning Techniques for the Performance Enhancement of Multiple Classifiers in the Detection of Cardiovascular Disease from PPG Signals

**DOI:** 10.3390/bioengineering10060678

**Published:** 2023-06-02

**Authors:** Sivamani Palanisamy, Harikumar Rajaguru

**Affiliations:** 1Department of Electronics and Communication Engineering, Jansons Institute of Technology, Coimbatore 641659, India; 2Department of Electronics and Communication Engineering, Bannari Amman Institute of Technology, Sathyamangalam 638402, India

**Keywords:** CVD, Hilbert transform, ABC PSO, harmonic search, nonlinear regression, principle component analysis, expectation maximization, Gaussian mixture model, Bayesian linear discriminant analysis classifier

## Abstract

Photoplethysmography (PPG) signals are widely used in clinical practice as a diagnostic tool since PPG is noninvasive and inexpensive. In this article, machine learning techniques were used to improve the performance of classifiers for the detection of cardiovascular disease (CVD) from PPG signals. PPG signals occupy a large amount of memory and, hence, the signals were dimensionally reduced in the initial stage. A total of 41 subjects from the Capno database were analyzed in this study, including 20 CVD cases and 21 normal subjects. PPG signals are sampled at 200 samples per second. Therefore, 144,000 samples per patient are available. Now, a one-second-long PPG signal is considered a segment. There are 720 PPG segments per patient. For a total of 41 subjects, 29,520 segments of PPG signals are analyzed in this study. Five dimensionality reduction techniques, such as heuristic- (ABC-PSO, cuckoo clusters, and dragonfly clusters) and transformation-based techniques (Hilbert transform and nonlinear regression) were used in this research. Twelve different classifiers, such as PCA, EM, logistic regression, GMM, BLDC, firefly clusters, harmonic search, detrend fluctuation analysis, PAC Bayesian learning, KNN-PAC Bayesian, softmax discriminant classifier, and detrend with SDC were utilized to detect CVD from dimensionally reduced PPG signals. The performance of the classifiers was assessed based on their metrics, such as accuracy, performance index, error rate, and a good detection rate. The Hilbert transform techniques with the harmonic search classifier outperformed all other classifiers, with an accuracy of 98.31% and a good detection rate of 96.55%.

## 1. Introduction

PPG has demonstrated that it is an effective device for the early diagnosis of cardiac disorders. Using PPG, blood volume fluctuations in tissues are continuously measured. PPG serves as a substantial promising technique for the thorough screening of cardiovascular diseases (CVDs). Blood circulation from the heart to the toes and fingertips is measured using PPG signals [1]. PPG sensors typically have an infrared-operating avalanche photo diode as a detector and a light-emitting diode (LED) as the source [2]. Both the European Committee for Standardization and the International Standards Organization (ISO) have recognized PPG as a standard noninvasive procedure for measuring and analyzing blood oxygen saturation levels. CVD leads to changes in the heart rate. PPG is skin-friendly, since it does not require direct skin-to-surface contact. Cardiac functions, such as blood flow, heart rate, and mean circulation time, are measured using PPG signals [3], since heart rate is associated with multiple physiological variables connected with hormonal and neuronal disturbances, pumping mechanisms, and a mean blood circulation time. A variation in the heart rate is viewed as a frequency shift over time and is specified as quasi-periodic in nature. A spectral analysis of PPG is used to determine the Eigen structure approaches as established by Al-Fahoum et al. [4]. The methods described in [4] are more appropriate to localize the spectral content of the PPG signal, which will identify the variations in the heart rate. Sukor et al. [5] discussed the presence and removal of signal artifacts in PPG signals using the waveform morphology method and achieved an accuracy of 83 ± 11% in the detection of CVD. CVDs are becoming one of the top causes of death worldwide. Approximately 16.7 million deaths worldwide, or 29.2% of all deaths, have occurred as a result of the different types of CVDs over the past ten years. All things considered, doctors and wellbeing experts advocate for the early discovery and treatment of potential side effects to guarantee the shirking of an out-and-out heart-related disease [6]. The main disadvantage of PPG signals is that they are highly corrupted by noise components, such as skin artifacts and motion artifacts. Hence, the preprocessing of PPG signals to attain cleaning signals is, itself, a broad area of research. Tun [7] designed an FIR filter for a heart rate detection system, which helps to minimize motion artifacts in a light-based measuring system. The reduction in artifacts in PPG signals using an AS-LMS adaptive filter was discussed by Ram et al. [8], and Luke et al. [9] presented a methodology using efficient signal processing algorithms for artifact removal in PPG signals and attained an SNR value of 41.52 db. From the analysis of PPG signals, the common parameters extracted are the pulse transit time, heart rate, stiffness index, and pulse wave velocity. Large artery stiffness and pulse width, pulse interval, systolic amplitude, augmentation index, and peak-to-peak interval are certain characteristics extracted from PPG signals [10]. Shintomi et al. [11] used mobile and wearable sensor equipment to measure the effectiveness of the heartbeat error and compensation strategies on heart rate variability (HRV). PPG is frequently used instead of an electrocardiogram (ECG) to assess heart rate in wearable devices. However, there are inherent differences between PPG and ECG due to the fact that PPG is affected by body motions, vascular stiffness, and blood pressure variations. Using ECG and PPG readings obtained from 28 people, the methods described in [11] determined how these errors affected the analysis of HRV. The assessment’s findings demonstrate that the error compensation method enhances the precision of HRV analysis in both the time and frequency domains as well as in nonlinear analysis. When compared to recent ECG observing systems, PPG signal measurements are more accessible and require less hardware for signal acquisition [12]. PPG does not require a reference signal, making it possible to integrate PPG sensors with wristbands. As a result, it can be employed in a variety of studies for the investigation and diagnosis of CVDs [13]. Moshawrab et al. [14] discussed how CVDs can be detected and predicted using smart wearable devices. In addition, the authors indicated that a review of the evaluation of the development and usage of smart wearable devices for the management of CVD demonstrates the high efficacy of such smart wearable devices.

The main objective of this research was to enhance the performance of multiple classifiers in the detection of cardiovascular disease (CVD) from PPG signals. The PPG signals obtained from the Capno database consisted of various features and occupied a large amount of memory, so it was necessary to extract the useful features by dimensionality reduction techniques. The main contributions of this work are that, by utilizing heuristic- and transformation-based dimensionality reduction techniques, the dimensions of the PPG data were reduced. The dimensionality-reduced PPG data were processed with twelve different classifiers, such as PCA, EM, logistic regression, GMM, Bayesian LDC, firefly, harmonic search, DFA, PAC Bayesian learning, KNN-PAC Bayesian, SDC, and detrend SDC. Machine learning techniques were used to improve the classification performance of the classifiers in the detection of CVDs from PPG signal.

The performance of the classifiers in terms of the classification accuracy, GDR, and computational complexity were examined and analyzed as the outcomes of this research. This article is organized as follows: The introduction is presented in Section 1, followed by the methodology in Section 2. Section 3 discusses the five techniques for reducing dimensionality. The twelve classifiers used to classify the normal and CVD segments of the PPG signals are discussed in Section 4. In Section 5, the obtained results are discussed. In Section 6, the article is concluded.

### Review of Previous Work

Certain works associated with PPG signal exploration and their use in various biomedical areas are provided below. For approximately three decades, research on PPG signals with respect to CVDs has been very interesting, and many research outcomes have been reported. Allen [15] examined PPG and how it could be used in clinical physiological measurements through the clinical monitoring of the human body, autonomic function, and vascular assessment. Sunil Kumar and Harikumar [16] used parameters, such as independent component analysis (ICA), principal component analysis (PCA), entropy, and mutual information (MI) to analyze the PPG signals. PPG has proven to be the most capable method for the early screening of heart-related diseases. Almarshad et al. [17] disclosed the diagnostic properties of PPG signals in addition to their prospective clinical applications in healthcare and assessed the possible effects of PPG signals on the screening, monitoring, diagnosis, and fitness of inpatients and outpatients. Yousefi et al. [18] developed a technique to automatically detect premature ventricular contraction (PVC) from the extracted features of PPG signals using higher-order statistics (HOS) and the chaotic method for the KNN classifier, with a classification accuracy of 95.5%. Cardiac health supervision based on PPG through smart phones or wearable sensors has the potential to attain high accuracy with fewer false alarms, as mentioned by Ukil et al. [19]. With the help of the ensemble method, Almanifi et al. [20] used a computer-vision approach to find human activities using PPG signals. The accuracy of the PPG was 88.91%, which shows that wrist PPG data could be used with the ensemble method in human activity recognition (HAR) to make accurate detections. The time domain analysis of a PPG signal and its second derivative was performed by Paradkar et al. [21] to extract the features from the PPG signal, and these extracted features were classified by the SVM classifier to detect coronary artery disease (CAD) with a sensitivity of 85% and a specificity of 78%. Neha et al. [22] investigated the detection of arrhythmias using PPG signals. In this analysis, a low-pass Butterworth filtering method was used to remove artifacts, and the extracted features were applied to various machine learning algorithms to classify normal and abnormal pulses. The results show that the SVM classifier had a high accuracy of 97.674% in identifying arrhythmia pulses. Prabhakar et al. [23] analyzed metaheuristic-based optimization algorithms as dimensionality reduction techniques and then applied various classifiers to these dimensionally reduced values for the classification of CVD. The results in [23] show that, for chi-squared probability density function (PDF) optimized values, the artificial neural network (ANN) classifier attained a maximum classification accuracy of 99.48% for normal subjects, and the logistic regression classifier produced a maximum classification accuracy of 99.48% for CVD cases. In order to analyze CVD patients utilizing PPG signals, Sadad et al. [24] applied various machine learning techniques and a deep learning model to create a system that helps doctors with continuous monitoring, and they obtained an accuracy of 99.5%.

In this article, a PPG signal was dimensionally reduced using five dimensionality reduction techniques, namely the Hilbert transform (HT), nonlinear regression (NLR), ABC PSO, cuckoo search, and dragonfly methods. These dimensionally reduced PPG signals were further classified as CVD or normal using twelve classifiers. Principle component analysis (PCA), expectation maximization (EM), logistic regression (LR), Gaussian mixture model (GMM), Bayesian linear discriminant analysis classifier (BLDC), firefly clusters, harmonic search, detrend fluctuation analysis (DFA), PAC Bayesian learning, KNN-PAC Bayesian, softmax discriminant classifier (SDC), and detrend with SDC were the classifiers used for classification purposes.

## 2. Methodology

The PPG data recordings with various waveform morphologies utilized in this work were obtained from the IEEE TMBE pulse oximeter standard CapnoBase database dataset, which is available online [25]. The dataset comprises raw PPG signal recordings with a duration of 8 min. This database consists of annotated respiratory signals, such as pressure, respiratory flow, and inhaled and exhaled carbon dioxide (capnogram). All 41 records were considered in this investigation. Twenty-one of the forty-one cases are normal, while the other twenty have cardiovascular disease. The dataset for the PPG signals was sampled at a rate of 300 Hz, and 144,000 samples of data were extracted from the PPG, with 720 segments overall. Each of these segments had 200 samples at equal intervals.

Independent component analysis (ICA) was used to remove the noise components in the PPG signals. The classification of a PPG signal involved two steps: First, dimensionality reduction (DR) was achieved with the help of heuristic- and transformation-based techniques. Second, these dimensionally reduced values were classified using various classifiers to detect whether the corresponding PPG signal was associated with a person with CVD or a normal subject. After the implementation of the dimensionality reduction techniques, the original PPG samples of a patient (200 × 720) were reduced to 100 × 720. These dimensionally reduced samples (100 × 720) were input into the classifiers for further classification. The organization of the CVD detection from the PPG signals is depicted in Figure 1.

## 3. Dimensionality Reduction Methods

Dimensionality reduction (DR) is a preprocessing technique used to eliminate irrelevant data and redundant features in order to reduce the training time of PPG signals [26]. All machine learning techniques and models become increasingly challenging as the dimensions of the input dataset increase. Dimensions are a problem in PPG signals with large amounts of data. As the number of features increases, the number of samples also increases proportionately, and the chances of overflow also increase. When a machine learning model is trained on large datasets, it becomes superabundant and results in a mediocre performance. As a result, it is necessary to decrease the number of features, which can be accomplished by dimensionality reduction. In this way, the DR technique is used to prevent data overfitting and select the most informative characteristics for classification purposes [27]. In this research, five dimensionality reduction techniques, divided into two categories, were utilized. First, transformation-based optimization techniques include Hilbert transformation and NLR optimization. Second, heuristic-based optimization includes ABC PSO, cuckoo search, and dragonfly optimization. These methods are discussed in the following.

### 3.1. Hilbert Transform

The Hilbert transform (HT) is a mathematical operation that is used to obtain the analytical representation of a real-valued signal.

The Hilbert transform [28] of signal *y*(*t*) is given by
(1)$$\hat{y}\left(t\right)=H\left[y\left(t\right)\right]=\frac{1}{\pi }\int_{-\infty }^{\infty }\frac{y\left(\tau \right)}{t-\tau }d\tau \;$$

It can be observed from the above equation that this transformation has no effect on the independent variable; therefore, the output $\hat{y}\left(t\right)$ is also a function that changes over time. Furthermore, the output $\hat{y}\left(t\right)$ is a linear function of input $y\left(t\right).$ It is produced by applying convolution to $y\left(t\right)$ with ${\left(\pi t\right)}^{-1}$, as shown in the equation below:(2)$$\hat{y}\left(t\right)=\frac{1}{\pi t}*y\left(t\right)$$

By applying Fourier transform (FT) to the above equation, we obtain
(3)$$F\left\{\hat{y}\left(t\right)\right\}=\frac{1}{\pi }F\left\{\frac{1}{t}\right\}F\left\{y\left(t\right)\right\}$$

A phase shift of −90 degrees is produced on all positive frequency components of the real-valued signal *y*(*t*) and +90 degrees on all negative frequency components when a Hilbert transform is applied to *y*(*t*). The domain of the signal is not changed by HT. When a Hilbert transform is applied to two different signals with the same amplitude but different phases, the magnitude spectrum is the same because the transform does not change the magnitude spectrum but changes the phase spectrum. This small phase response is obtained from spectral analysis using the Hilbert transform. In signal processing, the Hilbert transform is frequently used to generate the analytical representation of the real-valued signal *y*(*t*). All Fourier transformable signals are Hilbert transformable [29].

### 3.2. Nonlinear Regression

Nonlinear regression (NLR) is a statistical technique that involves developing a regression model to represent a nonlinear relationship between independent and dependent variables. The fundamental concept of linear regression and nonlinear regression is the same, that is, to connect a response, R, to a set of predictors, $Z={\left(z_{1},z_{2}\ldots \ldots .z_{n}\right)}^{T}$. The prediction equation for nonlinear regression varies nonlinearly on one or more unidentified parameters. Typically, nonlinear regression occurs when there is a specific functional shape in the relationship between the predictors and the response. The main goal of this model is to provide low sums of squares. The sum of squares parameter is connected to the number of observations that deviate from the dataset’s mean value. The variance between the mean and the individual points in the dataset is used to estimate the mean or average parameter. The collected variants are squared before being added together in the final step. The objective best matches the dataset points if the sum of the squared variations is found to be low. The least squares method, the equations of which comprise nonlinear elements, is used to compute the parameters in a nonlinear model. The steepest descent, Taylor series, and Levenberg–Marquardt methods can be used to solve these kinds of equations. The least squares in a nonlinear model is calculated by applying the Levenberg–Marquardt algorithm, which is its most extensive use. The advantages of this strategy include the optimum feature selection and deserving model convergence through iterations.

The structure of a nonlinear regression model as shown in [30] is
(4)$$R_{i}=f\left(z_{i},\varphi \right)+e_{i}\;\mathrm{for}\;i=1,2,\ldots ..n$$
where $R_{i}$ are responses; f is a known function of the covariate vector $Z_{i}={\left(z_{i1},z_{i2}\ldots \ldots .z_{in}\right)}^{T}$ and the vector parameter $\varphi ={\left(\varphi_{1},\varphi_{2}\ldots \ldots .\varphi_{n}\right)}^{T}$; and $e_{i}$ represents the random error values. Typically, it is assumed that random errors have an uncorrelated mean of zero and constant variance.

The formula for calculating the residual sum of squares is expressed as
(5)$$S\left(\varphi \right)=\sum_{i=1}^{n}{\left[R_{i}-f\left(z,\varphi \right)\right]}^{2}$$

### 3.3. ABC-PSO

ABC-PSO stands for artificial bee colony–particle swarm optimization. It is a hybrid optimization algorithm that combines the artificial bee colony (ABC) algorithm and the particle swarm optimization (PSO) algorithm to enhance the search capability and convergence speed of the optimization process. The ABC method has the advantages of simplicity, flexibility, and fast achievement of good results for multidimensional datasets. It is obvious that, in order to locate the locations of new food sources, the ABC algorithm does not necessarily need to apply the population’s best global solution. In the meantime, it may be concluded that PSO particles may not be able to escape from the local minima by a performing random search as scout bees in ABC. In addition, the update equation in ABC updates a single variable instead of all variables as in PSO. In order to achieve the best results, the ABC algorithm has been combined with the PSO search algorithm. In ABC-PSO, three ABC phases are used, and for the employed bee phase, it uses velocity and the PSO’s method of locating new food sources. The best location currently visited by the person is updated after the position of a new food supply is updated. The food source’s trial counter is reset if the current best position is altered; otherwise, its value is raised by 1. Onlooker bees memorize their positions throughout the employed bee phase and search for new food sources based on their knowledge of the best food source locations that the employed bees have visited. The new candidate solution “*z*” is generated by utilizing the ABC update equation, as follows:(6)$$z_{mn}=\left\{\begin{array}{c} pbest_{mn}\;\mathrm{if}\;n\neq k\\ pbest_{mk}+\emptyset_{mk}.\left(pbest_{mk}-pbest_{lm}\right),\;\mathrm{if}\;n=k \end{array}\right.$$
where $z_{mn}$ is the mth dimension of the nth employed bee selected; k is the random index; l is the index of a randomly selected individual; and $\emptyset_{mk}$ is a random number between −1 and 1. The food source trial counter will reset if the new food source location has a better value; otherwise, it is raised by 1. The scout bee phase follows the ABC algorithm [31].

ABC-PSO Algorithm:1.Initialization of the swarm.2.Velocity and position of the particle are updated by performing the employed bee phase.3.The local best position of a particle is updated by finding its new position by performing the onlooker bee phase.4.If the highest value of the trial counter for any food source is higher than the limit, a scout bee will search for a new food source site.5.At this point, instead of using scout bees, the PSO algorithm is used to look for new food sources.6.Particles with random placements are used to initialize the population of new food sources.7.The fitness value is determined for all particles for the specific objective function.8.The fitness function is used to select the optimal set of features. The expression for the fitness function is as follows:
(7)$$F=a\varphi_{k}+b\frac{\left|T\right|-\left|K\right|}{\left|T\right|}$$
where $\varphi_{k}$ is the classifier performance in subset K; b is the feature subset length; T is the total number of features; and a is the classification quality.9.The number of particles that are currently present is set as pbest.10.A new set of particles is created by adding velocity to the initial particle, and a fitness value is calculated for the same.11.A new pbest is discovered between the two particle sets by comparing the fitness values of each particle to one another.12.The least fitness value is determined by comparing the two sets of particles, and the corresponding particle is then referred to as the Gbest.13.Simultaneously, in the next iteration, the update in the velocity ($v^{q+1})$ and position $\left(x^{q+1}\right)\;$ is conducted as follows:(8)$$v^{q+1}=v^{q}+a\left(pbest-x^{q}\right)+b\left(Gbest-x^{q}\right)$$
(9)$$x^{q+1}=x^{q}+v^{q+1}$$The maximum step size that a particle can take in each iteration is influenced by the acceleration coefficients a and b.14.The PSO iterations are continued until convergence is reached.15.The finest food source is identified and remembered.16.The process can be performed as many times as necessary to fully satisfy the stopping criteria.

The main goal of this combined hybridization method is to combine the elements of ABC and PSO so that separate problems may be readily addressed by ABC and PSO simultaneously addressing rotationally invariant behavior.

### 3.4. Cuckoo Search

The cuckoo search (CS) algorithm is a nature-inspired metaheuristic optimization algorithm, which is based on the brood parasitism of some cuckoo species, along with Levy flight random walks established by Xin-She Yang and Suash Deb [32]. This optimization method depends on the incubate parasitism activities of certain cuckoo types along with the behavior of the Levy flights of certain fruit flies and birds. The three common rules of the CS algorithm can be given as follows:At a certain time, every cuckoo bird lays one egg and dumps it in an arbitrary selected host nest.The subsequent generation will carry the top-quality eggs, which are present in the best host nests.There are only fixed quantities of host nests available, and a host bird can realize a cuckoo’s egg with a probability of $P_{a}\in \left[0\;1\right]$. In this instance, a cuckoo’s egg in the host nest may be thrown away by the host bird or it abandons the nest and creates an entirely new nest in a different location.

The fundamental steps of the cuckoo search algorithm can be summed up as follows using the above three rules.

Create a population of N host nests at the beginning.Randomly select the host nest X.Lay the egg in the selected host nest X.Compare the fitness of the cuckoo’s egg with the host egg’s fitness.If the fitness of the cuckoo’s egg is better than the host egg’s fitness, replace the egg in nest X with the cuckoo’s egg.Abandon the nest if the host bird notices and build a new one.Repeat steps 2–6 until the termination criteria are met.

Levy flight is the term used to describe the random flight patterns that birds use to find their next position, $z_{i}^{\left(t+1\right)}$, based on the present position, $z_{i}^{\left(t\right)}$. Levy flights are used to build new hosts in new places. Consider cuckoo “*i*”; a Levy flight is performed to produce the new results $z^{t+1}$:(10)$$z_{i}^{\left(t+1\right)}=z_{i}^{t}+\beta ⊕\mathrm{levy}\left(\Upsilon \right)$$
where ⊕ denotes the entry-wise multiplication, and β>0 is a scaling factor that denotes the step size. Here, the β value is considered the one to optimize. A random walk is provided by the Levy flight, and random step sizes are calculated from the Levy distributions as follows:(11)$$\mathrm{Levy}\sim v=t^{-\Upsilon }(1<\Upsilon \leq 3)$$

This has an infinite variance and mean. The symbol “~” indicates that the numbers being generated are pseudorandom and are drawn from a probability distribution. The consecutive jumps of a CS are basically a random walk procedure that follows a step-length distribution of a power-law through a heavy tail. Utilizing the Levy walk approach, several new solutions around the current best solution should be produced by wide-area randomization. The locations of the new solutions should be far from the current best solution. This will prevent the system from becoming stuck at a local optimum.

### 3.5. Dragonfly

The dragonfly algorithm (DA) is a swarm intelligence algorithm, and it is inspired by the dynamic and static swarming behaviors of dragonflies. The dragonfly algorithm is a modern heuristic optimization technique created by Mirjalili in 2016 [33]. The static and dynamic swarming behaviors of dragonflies in nature serve as the primary source of inspiration for the dragonfly algorithm. In the dynamic or exploitation phase, a huge number of dragonflies form swarms and travel over long distances in one particular direction to distract their enemies. In the static or exploration phase, dragonflies form groups and move frontward and backward in a small zone for hunting and attract their prey. The five fundamental principles of DA are separation, alignment, cohesiveness, attraction, and diversion. In the following equations, Q and $Q_{j}$ denote the current and jth  positions of the individual dragon flies, respectively, and the total number of neighboring flies is denoted by K.
1.Separation: This indicates the static avoidance of flies colliding with other flies in the area. It is calculated as
(12)$$S_{i}=-\sum_{j=1}^{K}Q-Q_{j}$$
where $S_{i}$ denotes the separation motion of the ith individual.2.Alignment: This signifies the velocity matching among individual flies within the same group. It is denoted as
(13)$$A_{i}=\frac{\sum_{j=1}^{K}V_{j}}{K}$$
where $V_{j}$ denotes the velocity of the jth individual.3.Cohesiveness: This denotes the tendency of individual flies to move to the center of swarms. The estimation of this is given by
(14)$$C_{i}=\frac{\sum_{j=1}^{k}Q_{j}}{K}-Q$$4.Attraction towards the nourishment source is estimated as
(15)$$F_{i}=Q^{+}-Q$$
where $F_{i}$ denotes the nourishment source of the ith individual and $Q^{+}$ is the position of the nourishment source.5.Diversion: This represents the distance outwards to the enemy. It is calculated as
(16)$$E_{i}=Q^{-}+Q$$
where $E_{i}$ denotes the ith individual enemy’s position and $Q^{-}$ denotes the enemy’s position.

Within the search space, the locations of artificial dragonflies are updated by the step vector, ΔQ, and the current position vector, Q. The direction of the dragonfly’s movement is indicated by the step vector, ΔQ, and it is evaluated as
(17)$$\Delta Q_{i}^{t+1}=\left(sS_{i}+aA_{i}+cC_{i}+fF_{i}+eE_{i}\right)+\omega \Delta Q_{i}^{t}$$
where s, a, c, f, and e are the separation weight, alignment weight, cohesion weight, attraction weight, and enemy weight, respectively. The inertia weight is denoted by “ω” and “*t*” denotes the iteration number. The exploration and exploitation phases can be obtained by changing the weights.

At *t* + 1 iterations, the position of the ith dragonfly is calculated as follows:(18)$$Q_{i}^{t+1}=Q_{i}^{t}+\Delta Q_{i}^{t+1}$$

### 3.6. Statistical Analysis of Dimensionally Reduced PPG Signals

The dimensionally reduced PPG signals were analyzed through the extraction of statistical parameters and sample entropy for ascertaining the nonvariation in PPG signal characteristics. The statistical features [34], such as the mean, variance, skewness, kurtosis, Pearson correlation coefficient (PCC), and sample entropy [35], were extracted from the dimensionally reduced PPG samples among the CVD and normal classes. This reduced dataset provides the appropriate information from the above features.

Table 1 shows that the statistical analysis of the parameters was carried out with DR techniques for the PPG signals. It is observed from Table 1 that, for normal cases, lower mean values were obtained across the various optimization techniques. For cases of CVD, higher mean values, except for ABC-PSO and the negative mean, were obtained under dragonfly optimization. The skewness and kurtosis were highly skewed values for normal as well as CVD cases. It is inferred from Table 1 that the sample entropy values were the same across all classes, except for the NLR DR method in normal cases and the cuckoo search for the CVD cases. In addition, from Table 1, it is shown that the PCC values were low, which indicates that the optimized features were nonlinear and uncorrelated within the classes. Therefore, it is better to apply nonlinear classifiers to detect the CVD and normal segments of the PPG signals. If the CCA values are greater than 0.5, then there will be high correlation across the classes. From Table 1, it can be noticed that the Hilbert transform optimization was highly correlated across the classes. It also exhibits that the ABC-PSO optimization method was the least correlated among the classes.

Therefore, the above analyses of the dimensionally reduced PPG signals make a strong case for the usage of better classifiers. In order to identify the presence of nonlinearity in the dimensionally reduced signals, a normal probability plot for Hilbert transform-based dimensionally reduced values of the PPG signals in cases of CVD for Patient 2 is shown in Figure 2. From Figure 2, it can be observed that the normal plots exhibit the presence of nonlinearity and the overlapping of the Hilbert transformed values of the PPG signals.

Figure 3 shows a normal probability plot for the Hilbert transform-based dimensionally reduced values of the PPG signals in normal cases for Patient 30. The presence of outliers and nonlinear features of the Hilbert transform values can be observed. As a result, nonlinearity and outliers are preserved in the PPG signals of both classes after the DR methods.

## 4. Classifiers for Detection of CVD

Classifiers play a vital role in classifying data. An ideal classifier is one that provides high accuracy with a low error rate for a given computational complexity. The following sections of this paper discuss the classifiers that were used for this purpose.

### 4.1. PCA as a Classifier

Principal component analysis (PCA) is a technique that can be utilized for both data compression and classification purposes. The original m predictor variables are reduced to a smaller number of derived variables, n, by PCA. The derived variables are obtained by transforming the m original predictor variables, $Y=\left[y_{1},y_{2}\ldots .y_{m}\right]$, into a new variable set (the principal components) n, $Z=\left[z_{1},z_{2}\ldots .z_{n}\right]$. The new variables are linear combinations of the original variables. Mathematically, the eigenvalues and eigenvectors of the data covariance matrix are calculated to obtain principal components. The direction of the largest variation is identified from the eigenvector that has the largest eigenvalue [36].

### 4.2. Expectation Maximization as a Classifier

The expectation maximization (EM) algorithm is a method used to compute the maximum likelihood estimation in the presence of latent variables. Consider Z as the observed data, statistical parameters as φ, and the missing data as δ. The aim is to maximize the function.
(19)$$p\left(Z|\varphi \right)=\int p\left(Z,\left(\delta |\varphi \right)\right)d\delta$$

This equation cannot be systematically solved. It is assumed that the whole likelihood parameter or the posterior distribution $p\left(Z,\left(\delta |\varphi \right)\right)$ can be dealt easily with by applying the EM algorithm [37].

To reach convergence, this algorithm iterates between the E and M phases, as follows:E step (expectation): calculate the Q function:
(20)$$Q\;\left(\varphi |\varphi^{\left(t\right)}\right)=E_{\left(\delta |\varphi^{\left(t\right)}\right),Z}[\log p\left(Z,\left(\delta |\varphi \right)\right)]$$

M step (maximization): compute the maximum:
(21)$$\varphi^{\left(t+1\right)}=\arg\underset{\varphi }{\max}Q\left(\varphi |\varphi^{\left(t\right)}\right)\;$$
where “t“ denotes the iteration number. In the E-step, for each test point, the likelihood is computed from the individual cluster, and the calculation of the respective probability is carried out by assigning the test point to the corresponding cluster based on the maximum probability. All parameters are updated in the M step. This algorithm is repeated until it reaches convergence.

### 4.3. Logistic Regression as a Classifier

Logistic regression (LR) is a type of supervised machine learning algorithm that is utilized for predicting the probability of a target variable. The inputs are applied through a prediction function, which yields a probability value between 0 and 1, where 1 indicates CVD and 0 indicates normal [38]. To classify the positive and negative classes, a hypothesis, $H\theta \left(x\right)=\theta^{P}X$, is designed. The threshold value $H\theta \left(x\right)$ for the classifier is 0.5. If $H\theta \left(x\right)\geq 0.5$, then classify the values into the CVD class, and if $\;H\theta \left(x\right)<0.5$, then classify the values into the normal class. A classifier with at least a 50% predicted chance of classifying a CVD will be classified as class 1. Therefore, the threshold value $H\theta \left(x\right)$) for the logistic regression classifier was set to 0.5.

The LR function is given below:(22)$$H\theta \left(x\right)=g\left(\theta^{P}X\right)$$

### 4.4. Gaussian Mixture Model (GMM) as a Classifier

The Gaussian mixture model (GMM) is a machine learning technique utilized to classify data into different groups according to the probability distribution. Combinations of the number of Gaussian distributions are referred to as GMM. Given the data vector y, the GMM is defined as [39]
(23)$$p\left(y|\theta \right)=\sum_{q=1}^{Z}\pi_{q}M\left(y|\mu_{q},\sum_{q}\right)$$
where $\;\sum_{q},\;\mu_{q}$,$\pi_{q}$ are the covariance, mean, and mixture components of the GMM, respectively. In addition,
(24)$$\sum_{q}^{Z}\pi_{q}=1\;,\pi_{q}\geq 0\;\mathrm{Additionally},\;\theta =\left\{\mu_{1},\sum_{1},\pi_{1}\ldots \ldots \ldots \mu_{q},\sum_{q},\pi_{q}\right\}$$

The R-dimensional Gaussian distribution is represented by M:(25)$$M\left(y|\mu ,\sum \right)=\frac{1}{2\pi^{\frac{R}{2}}{\left|\sum \right|}^{\frac{1}{2}}}\exp\left[-\frac{1}{2}{\left(y-\mu \right)}^{T}\sum^{-1}\left(y-\mu \right)\right]$$

The EM algorithm is used to calculate the GMM’s parameters by applying the E and M steps.

E step: the posterior probability, $p_{iq}^{t}$, is evaluated at t iterations as
(26)$$p_{iq}^{t}=\frac{\pi_{q}^{t}p\left(y_{i}|\mu_{q}^{t},\sum_{q}^{t}\right)}{\sum_{q=1}^{z}\pi_{q}^{t}p\left(y_{i}|\mu_{q}^{t},\sum_{q}^{t}\right)}$$

M step: utilizing the probabilities evaluated from the E step, the parameters $\sum_{q},\;\mu_{q}$,$\pi_{q}$ are updated at t+1 iterations:(27)$$\pi_{q}^{t+1}=\frac{1}{N}\overset{N}{\underset{i=1}{\sum }}p_{iq}^{t}$$
(28)$$\mu_{q}^{t+1}=\frac{\sum_{i=1}^{N}p_{iq}^{t}y_{i}}{\sum_{i=1}^{N}p_{iq}^{t}}$$
(29)$$\sum_{q}^{t+1}=\frac{\sum_{q=1}^{z}p_{iq}^{t}\left(y_{i}-\mu_{q}^{t}\right){\left(y_{i}-\mu_{q}^{t}\right)}^{T}}{\sum_{i=1}^{N}p_{iq}^{t}}$$

To stabilize the parameters at a specific value, these two steps are repeated.

### 4.5. Bayesian Linear Discriminant Analysis as a Classifier

The Bayesian linear discriminant analysis classifier (BLDC) is a generative model that estimates the probability distribution of data for each class and uses the Bayes’ theorem to predict the class of new data. In order to maximize the class posterior probability, it is chosen from observation “k”, or in the case of two classes, x and  y. Choose class  x,  if $\;q_{x}\left(k\right)-q_{y}\left(k\right)\geq D$, where D denotes the decision threshold, and $q_{x}\left(k\right)$ is the discriminant function $,\;q_{x}\left(k\right)=\ln P\left(x|k\right)\;$ [40].

Assume that every class observation is taken from the multivariate normal distribution and that the covariance matrix is identical for all classes. Now, by applying the Bayes rule, the discriminant function is given as follows:(30)$$q_{x}\left(k\right)=-\frac{1}{2}{\left(k-\mu_{x}\right)}^{T}\sum^{-1}\left(k-\mu_{x}\right)+\ln P\left(x\right)$$
where $\mu_{x}$ is the mean feature vector for class x, Ʃ denotes the pooled covariance matrix for all classes, and $P\left(x\right)$ denotes class x’s prior probability. The prior probability of all classes is considered constant; then, the decision boundary, D, is given as follows:(31)$${\left(k-\mu_{y}\right)}^{T}\sum^{-1}\left(k-\mu_{y}\right)-{\left(k-\mu_{x}\right)}^{T}\sum^{-1}\left(k-\mu_{x}\right)\;$$

It is very clear that the mean vectors $\mu_{x}$ and $\mu_{y}$ are further away from each other in the feature space. If the term $\sum^{-1}\left(\mu_{x}-\mu_{y}\right)$ is larger, then the classes are more separable.

### 4.6. Firefly Algorithm as a Classifier

The firefly algorithm is a metaheuristic approach used for solving optimization problems, and it is inspired by the flashing patterns exhibited by fireflies, which was first developed by Yang [41]. The firefly algorithm employs three idealized rules:

All fireflies considered here are unisex in nature, and along these lines, one firefly will be attractive to other fireflies irrespective of sex.The attractive feature of a particular firefly varies with respect to its intelligence. Thus, for any two fireflies, the brighter firefly effectively pulls in the darker firefly. Assuming there are no fireflies brighter than a particular firefly, at that point, that specific firefly will move arbitrarily.When the distance from the firefly increases, the brightness or light intensity of a firefly will decrease because the light is captured as it passes through the air. Subsequently, the engaging quality or intelligence of a particular firefly, “s”, seen by firefly “t” is characterized as
(32)$$\alpha_{s}\left(r\right)=\alpha_{s}\left(0\right)e^{-\beta r^{2}}$$
where *β* is the light ingestion coefficient of a particular medium, $\alpha_{s}\left(0\right)$ signifies the brightness of firefly “s” at r=0, and *r* indicates the Euclidean distance between firefly “t”  and firefly “s”:(33)$$r=||z_{t}-z_{s}||=\sqrt{\sum_{i=1}^{d}{\left(z_{t}^{i}-z_{s}^{i}\right)}^{2}}$$
where $z_{t}\;$ and $z_{s}\;$ are the individual areas of fireflies “t”  and “s”, respectively. If firefly “s” is the brighter one, then its amount of attractiveness directs the movement of the specific firefly “t” as per the accompanying condition:(34)$$z_{t}=z_{t}+\left[\alpha_{s}\left(r\right)\right]\left(z_{s}-z_{t}\right)+\gamma \left(\mathrm{rand}\right)$$
where γ is the randomization parameter, and rand denotes a random number taken from a uniform distribution that lies in the range between −1 and +1, inclusively. Firefly “t”  can effectively move towards firefly “s” based on the second term in the above equation.

### 4.7. Harmonic Search as a Classifier

Harmony search (HS) is a music-based metaheuristic algorithm inspired by the evolution of music and the pursuit of the ideal harmony. Geem et al. [42] proposed that HS imitates the improvisational method used by musicians. The steps to be followed in the HS algorithm are:1.Problem Definition and HS Parameter InitializationAn unconstrained optimization problem is described as the minimization or maximization of the objective function, $f\left(Y\right)$, given as follows:$$Uy_{i}\leq y_{i}\;\leq Ly_{i}$$
where Y denotes the decision variable set; $y_{i}$ is the set of all possible values of every decision variable; and $Uy_{i}$ and $Ly_{i}$ represent the upper and lower bounds of the ith decision variable.2.Initialization of the Harmony MemoryIn this stage, the harmony memory (HM) is initialized. All decision variables in the HM are kept as matrices. The opening harmony memory is created from a uniform random distribution of values that are constrained by the parameters $Uy_{i}$ and $Ly_{i}$.3.Improvisation of a New HarmonyThe HM is utilized in this process to create a new harmony.4.Updating the Harmony MemoryThe HM is updated with the new harmony vector and the minimal harmony vector is deleted from the HM if the new improvised harmony vector is superior to the minimum harmony vector in the HM.5.Verification of the Terminating CriterionWhen the termination criterion is satisfied, the iterations are terminated. If not, steps 3 and 4 are repeated until the allotted number of iterations has been reached.

### 4.8. Detrend Fluctuation Analysis as a Classifier

Detrended fluctuation analysis (DFA) is a mathematical method used to analyze the presence of long-term correlations, or persistence, in a time series dataset. The main purpose of DFA is to scale long-range correlations in a time series. DFA is very similar to the Hurst exponent analysis, which is an enhancement of the standard fluctuation analysis [43]. DFA heavily relies on the random walk theory.

The cumulative profile, $M_{t}$, is obtained from the bounded time series, $m_{q},\;$ of length K, which is given by
(35)$$M_{t}=\sum_{q=1}^{t}\left(m_{q}-\langle m\rangle \right)$$

A local least squares straight-line fit is calculated by minimizing the squared errors inside each time window after dividing K into time windows, each with the length of *n* samples. The average fluctuation function is given by
(36)$$F\left(n\right)=\sqrt{\frac{1}{K}\sum_{q=1}^{K}{\left(M_{t}-N_{t}\right)}^{2}}$$
where $N_{t}$ is the piecewise sequence of the straight-line fits.

### 4.9. Probably Approximately Correct (PAC) Bayesian Learning Method as a Classifier

PAC Bayes is a generic framework to efficiently rethink generalization for numerous machine learning algorithms. It influences the flexibility of Bayesian learning and allows deriving new learning algorithms [44]. In PAC Bayesian learning theory, the hypothesis space is denoted as “*S*”, which is the variation between the empirical error and expected error of a hypothesis. The high-probability bounds of the deviation of the weighted averages of independent random variables are provided by the PAC Bayesian analysis of “s”, which is derived by the function $\theta \left(s\right).\;$ Prior distribution over the hypothesis space is denoted as Π, and the randomized classifier is defined as *γ*. Each time the game is played, the randomized classifier selects a hypothesis “s” from “*S*” in accordance with *γ* and uses it to predict the outcome of the subsequent sample.

### 4.10. KNN-PAC Bayesian Learning Method as a Classifier

KNN (K-nearest neighbors)-PAC (probably approximately correct)-Bayesian learning method is a machine learning algorithm used to make more accurate predictions by finding the nearest neighbors and updating the probability distribution of the data. The PAC Bayesian classifier is used to evaluate the divergence and training error of the finite data sample. For greater divergence, the risk factor will be high. The PAC Bayesian classifier output is fed to the KNN classifier to improve the accuracy of the classification. The KNN algorithm simply stores the dataset during the training phase and then classifies incoming data into a category that is very close to the previously stored dataset. A nonspecific sample can be classified considering the training data and samples. The selection of the K value is a critical stage in the KNN algorithm [45].

### 4.11. Softmax Discriminant Classifier (SDC) as a Classifier

The softmax discriminant classifier (SDC) is a supervised machine learning algorithm used for multiclass classification problems. It is based on the concept of the discriminant function, which maps input variables to a class label. SDC properly identifies the testing sample to which a particular class belongs by weighing the distance between the training and testing samples from that particular class [46]. Considering the training set $\;X=\left[X_{1},X_{2},\ldots .,X_{k}\right]\in R^{m\times n}$, chosen from “*k*” different classes, and $X_{k}=\lfloor X_{1}^{k},X_{2}^{k},\ldots .,X_{n_{k}}^{k}\rfloor \in R^{m\times n_{k}}$ represents $n_{k}$ samples from the $k^{th}$ class, where $\overset{k}{\underset{j=1}{\sum }}n_{j}=n$, and the testing samples are assumed to be $X\in R^{m\times 1}$.

Here, the SDC is defined as
(37)$$f\left(x\right)=\arg\;\underset{j}{\max}\;q_{x}^{j}$$
(38)$$f\left(x\right)=\arg\underset{j}{\max}\log\left(\sum_{i=1}^{n_{j}}\exp\left(-\lambda ||x-x_{i}^{j}{\left|\right|}_{2}\right)\right)$$
where $q_{x}^{j}\;,f\left(x\right)\;$ denotes the separation between the testing and jth class sample. A relative penalty cost is given when λ>0. Where x and $x_{i}$ have similar characteristics, if x belongs to the jth class, $||x-x_{i}^{j}{\left|\right|}_{2}$ is almost taken to zero and $q_{x}^{j}$ can asymptotically reach the maximum value.

### 4.12. Detrend with SDC as a Classifier

Detrend fluctuation analysis (DFA) with the softmax discriminant classifier (SDC) is a machine learning algorithm that combines the DFA and SDC techniques to classify time series data with long-range correlations by removing the trend and identifying the long-term correlations before classification. The correlation qualities of PPG signals can be determined over a long duration in DFA. SDC is used to determine and identify the class to which a particular test sample belongs.

## 5. Results and Discussion

This section explores the performances of different classifiers based on their benchmark parameters. A better classification accuracy with a lower error rate leads to a good classifier performance. Therefore, the classifiers were trained and tested for the dimensionally reduced values in the CapnoBase PPG signal dataset.

### 5.1. Training and Testing of the Classifiers

The training and testing of classifiers are very important steps for all of the classification processes. The training allows a classifier to learn the patterns associated with the given DR data. In this study, we chose 90% of the data for training and 10% for testing. The mean square error (MSE) was maintained as the stopping criteria for the training and testing of the classifiers. The mathematical expression for MSE is given below:(39)$$\mathrm{MSE}=\frac{1}{M}\sum_{i=1}^{M}{\left(O_{i}-T_{k}\right)}^{2}$$
where $O_{i}$ is the observed value at a definite time; $T_{k}\;$ indicates the model k’s target value, where “k” varies from 1 to 15; and M is assumed as 1000 and denotes the number of observations per case.

### 5.2. Selection of the Optimal Parameters for the Classifiers

Consider that the PPG dataset had two classes, namely CVD and normal, when determining the target values, and the target $T_{CVD}$ was carefully selected with higher values in the range from 0 to 1. The condition used for selecting $T_{CVD}$ is as follows:(40)$$\frac{1}{X}\sum_{i=1}^{X}\mu_{i}\leq \;T_{CVD}$$

The features of the total (X) CVD PPG data were normalized, and their mean is signified by $\mu_{i}$, as mentioned in Equation (40), which can be applied for the classification.

For normal subjects, the target $T_{Normal}$ with lower values between 0 and 1 was preferred when implementing the condition:(41)$$\frac{1}{Y}\sum_{j=1}^{Y}\mu_{j}\leq \;T_{Normal}$$

The features of the total (Y) normal PPG data were normalized, and their mean is signified by $\mu_{j}$, as mentioned in Equation (41), which can be applied for the classification.

$\mathrm{The}\;T_{CVD}$ value should be greater than that estimated for $\mu_{i}$ and $\mu_{j}$. It must be determined whether the difference between $T_{CVD}$ and $T_{Normal}$ is zero or greater than 0.5.
(42)$$∥T_{CVD}-T_{Normal}\geq 0.5∥$$

Depending on the condition given in (42), the $T_{CVD}$ and $T_{Normal}$ values were set as 0.85 and 0.1, respectively. The classifiers were trained with a 10-fold training and testing method, along with an MSE value of (10)^−5^ or a maximum operation of 1000, whichever was achieved earlier, as the stopping criterion. Table 2 demonstrates the selection of the optimal parameters for the classifiers.

Table 3 illustrates the analysis of the testing MSE values for the CVD and normal cases across the various classifiers with different DR techniques. It is perceived from Table 3 that, for the CVD cases, the ABC-PSO DR method with the DFA classifier resulted in the overall minimum MSE value of 4.00 × 10^−8^. The cuckoo search DR technique with the PCA classifier resulted in an overall maximum MSE of 6.60 × 10^−4^. Similarly, for normal classes, the overall minimum MSE of 9.00 × 10^−8^ was obtained when the Hilbert transform DR values were classified with the harmonic search classifier. The overall maximum MSE of 4.84 × 10^−4^ was obtained when the cuckoo search DR values were classified with the logistic regression classifier.

### 5.3. Performance Metrics of the Classifiers

In order to analyze the performance of the classifiers, the parameters, namely the performance index (PI), sensitivity, specificity, accuracy, good detection rate (GDR), and error rate, were calculated from the confusion matrix. Table 4 depicts the general confusion matrix for the detection of CVD.

True positive (TP): An output where the model accurately predicted the positive class, indicating that the person has cardiovascular disease.

True negative (TN): An output where the model accurately predicted the negative class, which shows that it is a healthy person.

False positive (FP): An output in which the positive class was incorrectly predicted by the model, which indicates that the healthy person is incorrectly classified as having CVD.

False negative (FN): An output in which the negative class was incorrectly predicted by the model, which indicates that a person with CVD is incorrectly classified as a healthy person.

PPG signals are sampled at 200 samples per second. Therefore, 144,000 samples per patient are available. There are 41 patients with 20 labeled has having CVD and 21 patients labeled as normal cases. Now, a one-second-long PPG signal is considered as a segment. Hence, there will be 720 such segments available per patient. The total number of segments for CVD cases is [20 × 720 = 14,400] and for normal cases is [21 × 720 = 15,120]. Therefore, the overall available segments for a total of 41 cases are 29,520 segments of one second duration only. The PPG signals are analyzed across the patients based on the signal segments. The beat to beat analysis is not included in this study. As a sample, the confusion matrix attained for the Hilbert transform DR method for different classifiers was undertaken, as shown in Table 5. As indicated in Table 5, the harmonic search classifier attained higher classification capability than the other eleven classifiers with low MSE values. At the same time, the firefly classifier well indented with more false negatives and less true positives categories.

These performance measures were calculated as follows:

The performance index (PI) is calculated as
(43)$$\mathrm{PI}=\left(\frac{\left(TP+TN\right)-FN-FP}{\left(TP+TN\right)}\right)\times 100$$

The sensitivity, specificity, accuracy, good detection rate (GDR), and error rate were calculated as shown below [19]:(44)$$\mathrm{Sensitivity}\;=\frac{TP}{TP+FN}\times 100$$
(45)$$\mathrm{Specificity}\;=\frac{TN}{TN+FP}\times 100$$
(46)$$\mathrm{Accuracy}\;=\frac{TP+TN}{TP+TN+FP+FN}\times 100$$

GDR: this represents the ability of a detector in fruitful detection, and it is given as
(47)$$\mathrm{GDR}=\left(\frac{\left[\left(TP+TN\right)-FP\right]}{TP+TN+FN}\right)\times 100$$
(48)$$\mathrm{Error}\;\mathrm{Rate}\;=\frac{FP+FN}{TP+TN+FP+FN}\times 100$$

As a sample for the classifier performance analysis, the cuckoo search DR method was undertaken, as shown in Table 6. As mentioned in Table 6, the harmonic search classifier performed better than all other classifiers in terms of the parametric values, such as an accuracy of 96.095%, performance index (PI) of 91.29%, sensitivity of 92.185%, specificity of 100%, highest GDR of 92.19%, and the lowest error rate of 7.81%. On the other hand, the firefly classifier showed the lowest performance with respect to all of the parametric values, such as an accuracy of 78.275%, a performance index (PI) of 20.76%, a sensitivity of 56.15%, a specificity of 100%, a GDR of 56.15%, and a high error rate of 43.855%. It is also observed from Table 6 that the GMM classifier reached a high sensitivity of 100% and ebbed at a low specificity of 84.38% due to the low number of true negative subjects. Even though there was 100% specificity for the PCA, logistic regression, BLDC, firefly, DFA, and PAC Bayesian learning classifiers, this did not assure good sensitivity values, except in the case of the harmonic classifier.

Table 7 exhibits the consolidated classifiers’ performance analysis across the different DR techniques. We can deduct from Table 7 that the Hilbert transformation DR approach with the harmonic search classifier retained its number one position with the highest PI of 96.485%, the highest accuracy of 98.31%, the lowest error rate of 3.38%, and the highest GDR of 96.55%, whereas the logistic regression classifier produced the lowest PI of 17.07%, the lowest accuracy of 77.38%, and the highest error rate of 45.245% under the Hilbert transformation DR technique. The lowest GDR of 40.755% was produced by the firefly classifier under the ABC-PSO DR method.

Figure 4 displays the performance analysis of the classifiers for the different DR techniques with respect to the error rate and GDR parameters. The harmonic search classifier achieved the highest GDR of 96.55%, with the lowest average error rate of 3.38% for the Hilbert transform DR technique, and the logistic regression classifier achieved the highest error rate of 45.245% for the Hilbert transform DR method, while the firefly classifier achieved the lowest GDR of 40.755% for the ABC-PSO DR technique.

Figure 5 demonstrates the performance analysis of the classifiers for the different DR techniques with respect to the accuracy. According to Figure 5, the harmonic search classifier clearly produced the highest accuracy of 98.31%, and the logistic regression classifier had the lowest accuracy of 77.38% for the Hilbert transformation dimensionality reduction method.

Next, we extensively examined the classifier accuracy as follows: for the PCA classifier, the higher accuracy of 87.45% was attained for the ABC-PSO DR method, and a lower accuracy of 80.36% was limited to the cuckoo search DR method. Figure 5 shows that the EM classifier maintained a high accuracy of 89.715% when using the cuckoo search DR method, while the NLR DR technique maintained a low accuracy of 82.255%. The high accuracy value for the logistic regression classifier was limited to 84.015% for the ABC-PSO DR method and a low accuracy of 77.38% for the Hilbert transformation DR technique. The GMM classifier placed a high accuracy of 95.12% with the dragonfly DR method and achieved a low accuracy of 86.79% with the NLR DR technique. The BLDC classifier secured a high accuracy of 90.63% with the cuckoo search DR method and retained a low accuracy of 80.21% with the ABC-PSO DR technique. The firefly classifier settled at a high accuracy of 94.145% with the NLR DR method and a low accuracy of 78.275% for the cuckoo search DR technique. For the harmonic search classifier, it showed a remarkable performance with a high accuracy of 98.31% with the Hilbert transformation DR method and maintained a low accuracy of 91.805% for the ABC-PSO DR technique. The harmonic search classifier maintained a high accuracy across the different DR methods, which was due to the better segregation and learning ability of the classifier. In the case of the DFA classifier, a high accuracy of 95.575% was achieved with the ABC-PSO DR method and a low accuracy of 88.38% was achieved with the NLR DR technique. The DFA classifier exhibited the second-best classification accuracy performance across the DR techniques. In the case of the PAC Bayesian learning classifier with the Hilbert transformation DR method, it retained a good accuracy of 83.525% and a low accuracy of 78.7% with the dragonfly DR method. The KNN-PAC Bayesian learning hybrid classifier reached a high accuracy of 89.26% for the Hilbert transformation DR method and maintained a low accuracy of 84.765% with the NLR DR method. For the SD classifier, it achieved a high accuracy of 94.99% with the cuckoo search DR technique, and it displayed a drastically low accuracy of 83.615% for the dragonfly DR technique. In the case of the DFA SDC hybrid classifier with the NLR DR method, it remained at a high accuracy at 92.585% and reached a low accuracy of 82.215% for the dragonfly DR method.

The robustness of the classifiers is reflected in the accuracy of the classifiers across the five dimensionality reduction techniques, namely Hilbert transform (HT), nonlinear regression (NLR), artificial bee colony–particle swarm optimization (ABC-PSO), cuckoo search, and dragonfly. All other classifiers, except the logistic regression classifier, settled at a higher accuracy of the maximum 80%, as shown in Table 7. This is due to the fact that the classifiers are trained and the optimal parameters for the classifiers are attained after the tuning process. The k-fold training and testing of the classifiers caused the classifiers to be more robust for the detection of CVD in the submitted PPG signal.

### 5.4. Summary of Previous Works on the Detection of CVD Classes

A summary of previous works on the detection of CVD classes is listed in Table 8. The time and frequency domain features, SVD, and stochastic features were extracted from the PPG signals, and these features were classified with various classifiers, such as ANN, KNN, ELM, GMM, softmax regression model, DNN, SDC, SVM, and harmonic search, to detect cases of CVD.

It is observed in Table 8 that Soltane et al. [47] proposed the artificial neural network (ANN) method to divide the PPG signal into two different classes. The input signal was smoothed to reduce the dimensionality, and the smoothing accuracy was used to explore the features in the multilayer feed-forward networks that were highly parallelized (MFN), and this achieved a classification rate for testing datasets of 94.7% and training datasets of 100%. Hosseini et al. [48] utilized finger PPG, a noninvasive optical signal collected before and after reactive hyperemia, to distinguish between people with various CVDs, with a maximum accuracy of 81.5% for the KNN classifier. Shobitha et al. [49] used the extreme learning machine (ELM), a supervised learning algorithm, to classify PPG signals as normal or affected by cardiovascular illness and compared its performance with backpropagation and support vector machine (SVM) techniques. These algorithms were validated by testing healthy and pathological signals from each of the 30 patients. In addition, with only five features as input, ELM produced the best accuracy, with a specificity of 90.33% and a sensitivity of 89.33%, and it also took less computational time to determine the risk of CVD. Prabhakar et al. [50] considered PPG signals obtained from a single patient. They extracted the statistical features, and the annotation of the PPG signals was conducted by using SVD. The annotated features of the class labels were verified and classified by the GMM and achieved an accuracy of 98.97%. This may be due to the smaller class vector size and overfitting condition for the GMM classifier. In the research work, heuristic- and transformation-based dimensionally reduced PPG data samples of 21 normal and 20 CVD cases were considered. The GMM classifier reached a maximum accuracy of 95.12% with the dragonfly dimensionality reduction technique. Based on patient diagnostic results for coronary heart disease, the classification and prediction models using deep neural networks (DNNs) were created and tested by Miao and Miao [51]. The created DNN learning model consisted of a classification model based on training data, and 303 clinical instances from patients with coronary heart disease at the Cleveland Clinic Foundation were used to create a prediction model for diagnosing new patient cases. The results of the tests indicate that the created classification and prediction model had an 83.67% diagnosis accuracy for heart disease. Hao et al. [52] proposed the softmax regression model, which employs neural networks for training and learning, and calculates the probability that reclassified data will fall into each category. This method classified CVD with an accuracy of 94.44%. Divya et al. [53] proposed a computer-aided diagnostic system that uses PPG signals to determine the different levels of CVD risk. From the PPG signals, statistical characteristics, wavelets, and singular value decomposition features were retrieved. By utilizing the SDC and GMM classifiers, the extracted feature vectors were classified to indicate the various risk levels of CVD, the results show that a classification accuracy of 97.88%, specificity of 99.09%, and a sensitivity of 97.24% were obtained by incorporating the SDC with value decomposition (SVD) and statistical features. In addition, a classification accuracy of 96.64%, specificity of 99.65%, and sensitivity of 93.80% were obtained by incorporating the GMM with SVD and statistical features. Prabhakar et al. [54] used a fuzzy-based approach to optimize the extracted parameters from PPG signals. The statistical features were extracted from the PPG signals, and fuzzy-based modeling was utilized to predict the CVD risk levels from the PPG signals. To optimize the fuzzy model levels, four types of optimization were performed. In order to produce the best results, the optimized values were categorized using the appropriate classifiers, and the support vector machine–radial basis function (SVM–RBF) classifier produced a maximum classification accuracy of 95.05% when the fuzzy model-based levels were optimized with animal migration optimization (AMO). A deep convolutional neural network was developed by Liu et al. [55] to classify multiple rhythms of 23,384 PPG waveforms from 45 patients and achieved an accuracy of 85%. Ihsan et al. [56] studied feature extraction algorithms, such as the respiratory rate (RR) interval, HRV features, and time domain features for detecting coronary heart disease using PPG and achieved an accuracy of 94.4% for HRV features using the decision tree classifier. Al Fahoum et al. [57] extracted the time domain features and health status information from PPG signals and applied feature selection-based classifiers in order to identify the difference between healthy persons and CVD patients. Seven distinct classifiers were utilized to classify the dataset and apply the feature selection. In the first stage, the naïve Bayes classifier achieved the highest accuracy of 94.44%, and in the second stage, an accuracy of 89.37% was attained. Rajaguru et al. [58] extracted the statistical features from the CapnoBase PPG signals of a single CVD patient, and the extracted features were classified with linear regression which produced a better accuracy of 65.85%.

In this research, the harmonic search classifier yielded the best classification accuracy of 98.31% for the HT DR values. The Hilbert transform is a linear operator that causes a 90-degree phase shift in a signal to obtain the desired separation, which is required in the exploration of phase in a harmonic search classifier. The harmonic search classifier is a pitch adjustment of the harmonics; therefore, more such phase exploration is possible to provide better classification. Hilbert transform segregates the signals at the first level itself, which reduces the burden of the classifiers; hence, the harmonic search classifier yielded a better classification accuracy. In order to identify a good classifier, the computational complexity performance measure plays a tradeoff role, as discussed below.

### 5.5. Computational Complexity Analysis of the Classifiers

The computational complexity may also be a performance metric for a classifier. Computational complexity is analyzed by utilizing an input size of *m*. If the size of the input is $\;O\left(1\right)$, the computational complexity will be very low. There will be an increase in computational complexity if there is an increase in the number of inputs. Computational complexity is denoted as O(logm) when it increases log *m* times with respect to the increase in *m*.

Table 9 indicates the computational complexity of the classifiers among the various dimensionality reduction techniques. Under the Hilbert transformation DR technique, the logistic regression and firefly classifiers had the lowest computational complexity of O(mlogm). The highest computational complexity of $O\left(m^{7}\right)$ was reached by the KNN-PAC Bayesian classifier with the ABC-PSO optimization technique. Even though ABC-PSO and DFA had the highest computational complexity of $O\left(m^{5}\right)$, a higher accuracy of 95.58% was exhibited. The higher accuracy of this classifier was due to the DFA’s characteristics feature. DFA identifies the peak value of the features, and the ABC-PSO will smooth the features and place them in the labeled classes without any outliers.

## 6. Conclusions

This study intended to detect cardiovascular disease (CVD) from PPG signals. The dimensionally reduced features obtained from the PPG signal were stored as datasets. Then, classifiers were used to detect CVD in the patients. The objective was to classify CVD with a high classification rate and a low rate of false positives and false negatives. Even though it is difficult to obtain perfect classification with classifiers, a compromise was made. As a high number of false positives decreases a classifier’s accuracy, therefore, a low number of false positives is the most important. The main limitation of this work is that the PAC Bayesian learning and logistic regression classifiers failed to achieve a higher classification accuracy across all five dimensionality reduction techniques, and thus the second-to-second detection of PPG classes will result in more false alarms. At the same time, 30 s segmented epochs of PPG signals were considered for the better classification accuracy of the classifiers. Under this circumstance, the classifiers will be overfitted with the training process and may end in higher accuracy. A compromise is made by taking the segment of a one-minute duration of raw PPG signals to attain a better classification accuracy. The results show that a high classification accuracy of 98.31% was attained when the Hilbert transform optimized values were classified with the harmonic search classifier, and a second highest accuracy of 97.79% was obtained when nonlinear regression optimized values were classified with the harmonic search classifier. A third highest accuracy of 96.095% was obtained when the cuckoo search optimized values were classified with the harmonic search classifier. It was also observed that the harmonic search classifier outperformed across all dimensionality reduction techniques. The convenience and real-time nature of a PPG-based method make it an attractive option for large-scale screening, which has the potential to be helpful in the long-term and real-time monitoring of CVD. PPG-based approaches could potentially be performed remotely without direct patient contact and with minimal patient training by wearable devices, such as fitness bands and smartwatches. As a result, the use of PPG-based methods could play a significant role in detecting CVD at an early stage and continuously measuring risk factors, leading to timely clinical evaluation. The further enhancement of the classifiers’ performance will be in the direction of the hyper-parameters’ selection through heuristic methods. The future research is toward CNNs and deep neural networks for the detection of CVD, with a minimum time lapse. Because CNNs are good at extracting features from PPG signals and identifying relevant patterns for CVD detection, deep neural networks can identify the most relevant risk factors and develop accurate models for CVD detection; by combining these two types of artificial intelligence, healthcare providers can more accurately diagnose and treat patients with CVD.

## Figures and Tables

**Figure 1 bioengineering-10-00678-f001:**
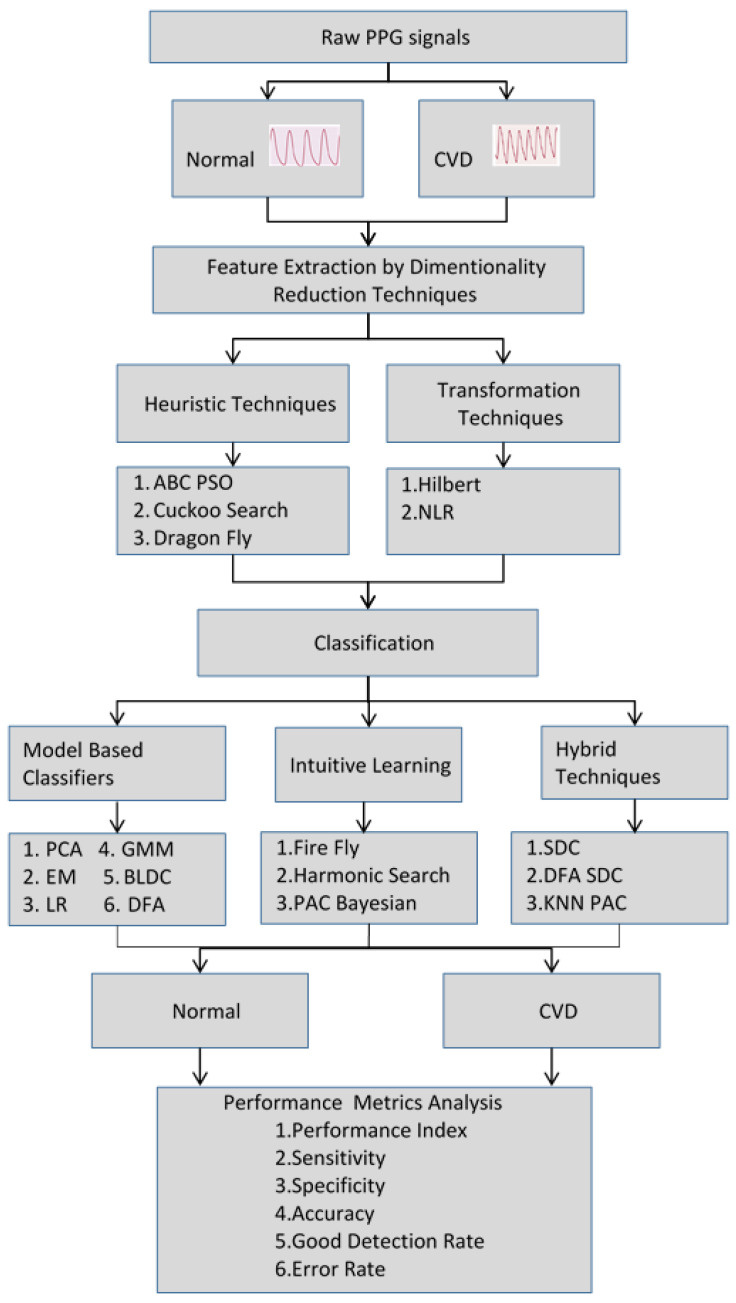
Organization of the CVD detection from PPG signals.

**Figure 2 bioengineering-10-00678-f002:**
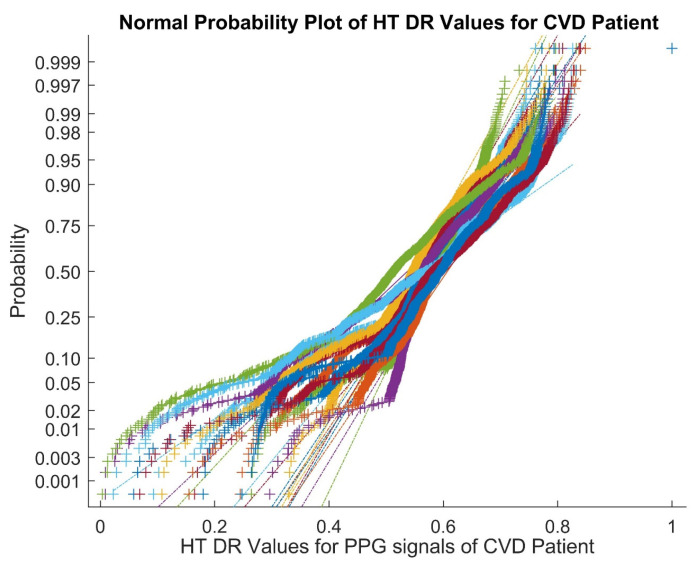
Normal probability plot for Hilbert transform-based dimensionally reduced values for the PPG signals in cases of CVD.

**Figure 3 bioengineering-10-00678-f003:**
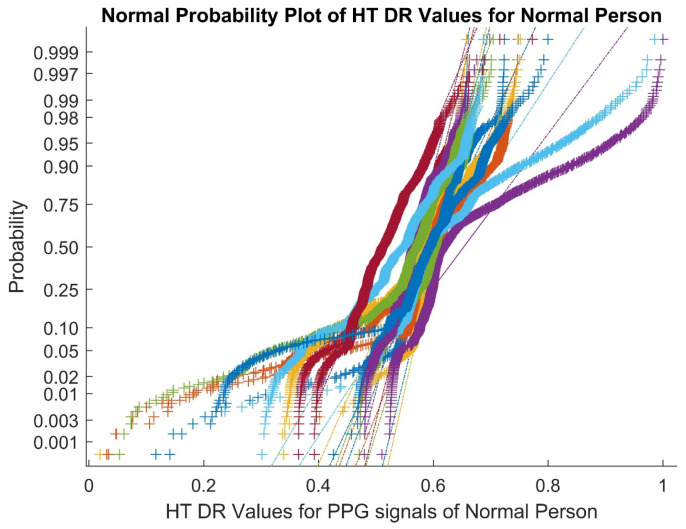
Normal probability plot for Hilbert transform-based dimensionally reduced values for PPG signals in normal cases.

**Figure 4 bioengineering-10-00678-f004:**
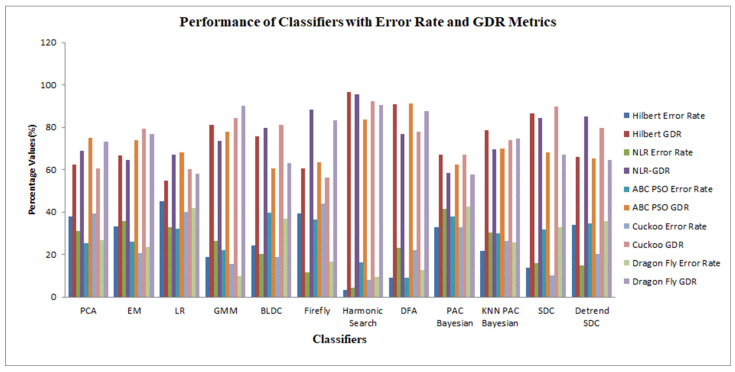
Performance of the classifiers in terms of the error rate and GDR metrics for the different dimensionality reduction methods.

**Figure 5 bioengineering-10-00678-f005:**
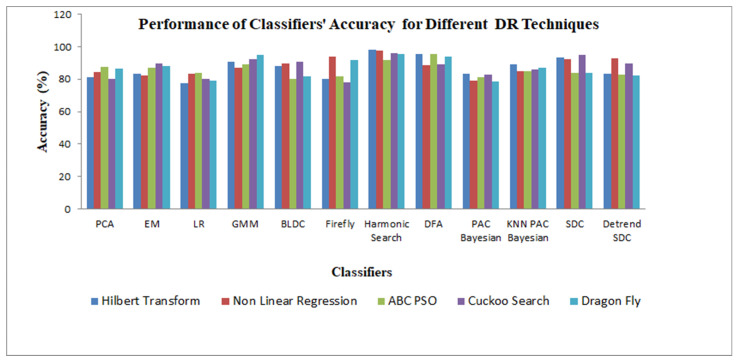
Performance of the classifiers’ accuracy for the different dimensionality reduction (DR) techniques.

**Table 1 bioengineering-10-00678-t001:** Average statistical parameters of different dimensionality reduction approaches for normal and CVD cases.

Statistical Parameters	Dimensionality Reduction Techniques									
	Hilbert Transform		NLR		ABC PSO		Cuckoo Search		Dragonfly	
	Normal	CVD	Normal	CVD	Normal	CVD	Normal	CVD	Normal	CVD
Mean	2.0611	5.457	2.126	117.469	0.08886	0.7939	0.5176	11.622	0.7803	−5.461
Variance	0.0706	1.162	23.056	225,124.7	0.0105	0.3412	0.0463	93.9237	411.1297	274.941
Skewness	−1.118	−0.8151	3.2789	5.769	−0.1046	−0.0888	0.1411	−0.1172	−0.0015	−0.0084
Kurtosis	5.626	1.638	14.59	46.05	0.2226	0.0991	−0.4971	−1.6769	−1.0065	−0.74424
PCC	0.3687	0.2332	0.01	0.0063	−0.0215	0.0134	0.2319	0.24	−0.1541	0.0775
Sample Entropy	9.7695	9.9328	6.9142	7.3959	9.9494	9.9473	9.9494	4.9919	9.9499	9.9522
CCA	0.5425		0.2198		0.1066		0.3674		0.4621	

**Table 2 bioengineering-10-00678-t002:** The selection of the optimal parameters for the classifiers.

Classifiers	Optimal Parameters of the Classifiers
Principal component analysis (PCA)	Decorrelated Eigen vector $w_{k}$, and a threshold value of 0.72 with the training of trial and error method with MSE of (10)^−5^ or a maximum iteration of 1000—whichever happens first.
Expectation maximization	Test point likelihood probability 0.15, cluster probability of 0.6, with a convergence rate of 0.6. Criterion: MSE
Logistic regression	Threshold $H\theta \left(x\right)=0.5$. Criterion: MSE
Gaussian mixture model (GMM)	Mean, covariance of the input samples and tuning parameter is EM steps. Criterion: MSE
Bayesian linear discriminant analysis (BLDC)	Prior probability *P(x):* 0.5, class mean $\mu_{x}=0.8\;$ and $\mu_{y}=0.1$. Criterion: MSE
Firefly algorithm classifier	γ = 0.1,${\;\alpha }_{s}\left(0\right)$ = 0.65 initial conditions, with an MSE of (10)^−5^ or maximum iteration of 1000—whichever happens first. Criterion: MSE
Harmonic search	Class harmony is fixed at target values for the classes 0.85 and 0.1. The upper and lower bounds are adjusted with a step size of ∆w of 0.004. The final harmony aggregation is attained with an MSE of (10)^−5^ or a maximum iteration of 1000—whichever happens first. Criterion: MSE
Detrend fluctuation analysis (DFA)	The initial values for K = 10000, M = 1000, and *n* = 100 are set to find *F*(*n*). Criterion: MSE
Probably approximately correct (PAC) Bayesian learning method	P(z), class probability: 0.5, class mean: 0.8,0.1; *γ* = 0.13, Criterion: MSE
KNN-PAC Bayesian learning method	Number of clusters = 2 with PAC Bayesian variables. Criterion: MSE
Softmax discriminant classifier (SDC)	λ = 0.5 along with mean of each class target values as 0.1 and 0.85.
Detrend with SDC	Cascaded condition of DFA with SDC classifiers with parameters as mentioned above.

**Table 3 bioengineering-10-00678-t003:** Analysis of the testing MSE for classifiers in CVD and normal cases.

Classifiers	Category	Hilbert Transform	NLR	ABC PSO	Cuckoo	Dragonfly
PCA	CVD	2.81 × 10^−5^	7.84 × 10^−6^	2.50 × 10^−7^	6.60 × 10^−4^	1.44 × 10^−6^
	Normal	3.06 × 10^−4^	2.79 × 10^−4^	2.40 × 10^−4^	3.72 × 10^−5^	1.21 × 10^−4^
EM	CVD	1.60 × 10^−5^	3.97 × 10^−5^	1.44 × 10^−4^	3.24 × 10^−6^	7.74 × 10^−5^
	Normal	2.28 × 10^−4^	6.24 × 10^−5^	8.10 × 10^−7^	3.03 × 10^−5^	2.50 × 10^−7^
Logistic regression	CVD	9.03 × 10^−5^	8.27 × 10^−5^	1.09 × 10^−5^	3.61 × 10^−5^	6.24 × 10^−5^
	Normal	3.23 × 10^−4^	1.84 × 10^−5^	2.56 × 10^−4^	4.84 × 10^−4^	1.33 × 10^−4^
GMM	CVD	1.44 × 10^−6^	4.41 × 10^−6^	6.40 × 10^−5^	4.84 × 10^−6^	6.76 × 10^−6^
	Normal	2.70 × 10^−5^	6.56 × 10^−5^	3.60 × 10^−7^	1.09 × 10^−5^	3.60 × 10^−7^
Bayesian LDC	CVD	2.92 × 10^−5^	2.50 × 10^−5^	5.49 × 10^−5^	1.00 × 10^−6^	2.22 × 10^−5^
	Normal	1.45 × 10^−5^	4.00 × 10^−6^	7.57 × 10^−5^	3.03 × 10^−5^	3.06 × 10^−4^
Firefly	CVD	4.41 × 10^−4^	6.25 × 10^−6^	8.84 × 10^−5^	6.40 × 10^−5^	8.41 × 10^−6^
	Normal	3.36 × 10^−5^	1.44 × 10^−6^	3.25 × 10^−5^	4.20 × 10^−4^	9.61 × 10^−6^
Harmonic search	CVD	1.60 × 10^−7^	4.90 × 10^−7^	3.61 × 10^−6^	3.24 × 10^−6^	5.29 × 10^−6^
	Normal	9.00 × 10^−8^	3.24 × 10^−6^	6.76 × 10^−6^	1.60 × 10^−7^	4.90 × 10^−7^
DFA (weighted)	CVD	1.60 × 10^−7^	2.56 × 10^−6^	4.00 × 10^−8^	1.30 × 10^−5^	2.50 × 10^−7^
	Normal	6.25 × 10^−6^	5.04 × 10^−5^	7.84 × 10^−6^	2.03 × 10^−5^	6.76 × 10^−6^
PAC Bayesian learning	CVD	4.90 × 10^−7^	3.35 × 10^−4^	3.06 × 10^−4^	1.68 × 10^−5^	5.78 × 10^−5^
	Normal	2.89 × 10^−4^	5.04 × 10^−5^	2.70 × 10^−5^	1.19 × 10^−4^	3.03 × 10^−4^
KNN-PAC Bayesian	CVD	7.84 × 10^−6^	8.41 × 10^−6^	6.25 × 10^−6^	1.00 × 10^−6^	8.41 × 10^−6^
	Normal	2.40 × 10^−5^	1.32 × 10^−4^	1.96 × 10^−4^	1.10 × 10^−4^	4.76 × 10^−5^
SDC	CVD	2.89 × 10^−6^	4.84 × 10^−6^	1.21 × 10^−4^	1.44 × 10^−6^	3.69 × 10^−4^
	Normal	8.41 × 10^−6^	1.15 × 10^−5^	1.31 × 10^−5^	2.57 × 10^−6^	1.16 × 10^−5^
Detrend SDC	CVD	2.40 × 10^−4^	8.10 × 10^−7^	3.06 × 10^−4^	1.02 × 10^−5^	4.41 × 10^−4^
	Normal	1.69 × 10^−5^	1.75 × 10^−5^	1.76 × 10^−5^	1.85 × 10^−5^	1.86 × 10^−5^

**Table 4 bioengineering-10-00678-t004:** Confusion matrix for the detection of CVD.

Actual Classification Class Output	Predicted Classification Class Output	
	CVD	Normal
CVD	TP	FN
Normal	FP	TN

**Table 5 bioengineering-10-00678-t005:** Confusion Matrix for Classifiers based on PPG Signal Segments for Hilbert Transform.

Classifiers	TP	TN	FP	FN
PCA	10,080	7920	7200	4320
EM	8640	7920	7200	5760
Logistic regression	11,520	7920	7200	2880
GMM	12,960	10,800	4320	1440
Bayesian LDC	10,080	12,240	2880	4320
Firefly	7200	10,800	4320	7200
Harmonic search	14,400	14,400	720	0
DFA (weighted)	13,680	12,960	2160	720
PAC Bayesian learning	11,520	7920	7200	2880
KNN-PAC Bayesian	7920	7920	7200	6480
SDC	12,960	12,960	2160	1440
Detrend SDC	7200	11,520	3600	7200

**Table 6 bioengineering-10-00678-t006:** Performance analysis of cuckoo search DR method for different classifiers.

Classifiers	PI (%)	Sensitivity (%)	Specificity (%)	Accuracy (%)	GDR (%)	Error Rate (%)
PCA	30.975	60.725	100	80.36	60.72	39.28
EM	72.6	94.01	85.42	89.715	73.425	20.57
Logistic regression	30.175	60.125	100	80.065	60.125	39.875
GMM	81.395	100	84.38	92.19	81.415	15.625
Bayesian LDC	75.205	81.255	100	90.63	81.255	18.745
Firefly	20.76	56.15	100	78.275	56.15	43.855
Harmonic search	91.29	92.185	100	96.095	92.19	7.81
DFA (weighted)	71.915	77.995	100	89	77.995	22.005
PAC Bayesian learning	48.605	67.1	100	82.84	67.085	32.915
KNN-PAC Bayesian	56.255	77.93	95.835	86.09	73.385	26.235
SDC	89.015	94.665	95.315	94.99	89.495	10.02
Detrend SDC	75.635	88.55	91.15	89.85	77.79	20.315

**Table 7 bioengineering-10-00678-t007:** Consolidated classifiers’ performance analysis across the different DR techniques.

Classifiers	Performance Metrics	Hilbert	NLR	ABC PSO	Cuckoo	Dragonfly
PCA	PI	35.69	47.195	55.145	30.975	55.215
	Error rate	37.78	31.07	25.1	39.28	26.69
	Accuracy	81.11	84.265	87.45	80.36	86.655
	GDR	62.22	68.935	74.9	60.72	73.31
EM	PI	47.1	44.285	55.41	72.6	60.925
	Error rate	33.35	35.545	26.11	20.57	23.3
	Accuracy	83.335	82.225	86.95	89.715	88.35
	GDR	44.485	64.455	55.57	73.425	61.05
Logistic regression	PI	17.07	48.955	46.195	30.175	27.255
	Error rate	45.245	32.86	31.98	39.875	41.93
	Accuracy	77.38	83.575	84.015	80.065	79.035
	GDR	54.755	51.13	66.125	60.125	45.555
GMM	PI	75.445	59.205	64.685	81.395	88.815
	Error rate	18.68	26.43	22.13	15.625	9.76
	Accuracy	90.66	86.79	88.935	92.19	95.12
	GDR	81.32	72.615	64.905	81.415	88.945
Bayesian LDC	PI	68.635	73.285	34.145	75.205	37.65
	Error ate	24.345	20.315	39.58	18.745	36.74
	Accuracy	87.83	89.845	80.21	90.63	81.63
	GDR	75.655	78.795	60.42	81.255	63.26
Firefly	PI	31.72	86.855	40.715	20.76	80.075
	Error rate	39.23	11.715	36.525	43.855	16.595
	Accuracy	80.39	94.145	81.74	78.275	91.7
	GDR	60.77	87.04	40.755	56.15	81.85
Harmonic search	PI	96.485	95.35	82.65	91.29	89.405
	Error rate	3.38	4.425	16.405	7.81	9.375
	Accuracy	98.31	97.79	91.805	96.095	95.315
	GDR	96.55	95.575	83.595	92.19	90.625
Detrend fluctuation analysis (weighted)	PI	89.57	66.12	89.65	71.915	87.29
	Error rate	9.11	23.235	8.85	22.005	12.495
	Accuracy	95.445	88.38	95.575	89	93.76
	GDR	90.89	76.765	91.15	77.995	87.505
PAC Bayesian learning	PI	44.95	26.885	35.82	48.605	25.285
	Error rate	32.96	41.635	37.715	32.915	42.39
	Accuracy	83.525	79.22	81.14	82.84	78.7
	GDR	64.74	58.365	62.285	67.085	46.055
KNN-PAC Bayesian	PI	71.875	49.755	50.005	56.255	63.69
	Error rate	21.48	30.47	29.95	26.235	25.45
	Accuracy	89.26	84.765	85.025	86.09	87.275
	GDR	78.52	69.535	70.05	73.385	74.55
SDC	PI	84.215	81.175	48.41	89.015	43.41
	Error rate	13.535	15.755	31.77	10.02	32.915
	Accuracy	93.23	92.125	84.115	94.99	83.615
	GDR	86.465	83.23	68.23	89.495	67.09
Detrend SDC	PI	45.635	83.395	42.93	75.635	39.775
	Error rate	33.825	14.84	34.655	20.315	35.585
	Accuracy	83.095	92.585	82.675	89.85	82.215
	GDR	66.175	84.875	65.345	77.79	64.42

**Table 8 bioengineering-10-00678-t008:** Summary of previous works on the detection of CVD classes.

Sl.no	Authors	Features	Classifier	Accuracy (%)
1	Soltane et al. [47] 2004	Time and frequency domain features	Artificial neural network	94.70%
2	Hosseini et al. [48] 2015	Time domain features	K-nearest neighbor	81.50%
3	Shobita et al. [49] 2016	Time domain features	Extreme learning machine	82.50%
4	Prabhakaret al. [50] 2017	Statistical features +SVD	GMM	98.97%
5	Miao and Miao [51] 2018	Time domain features	Deep neural networks	83.67%
6	Hao et al. [52] 2018	Statistical features	Softmax regression model	94.44%
7	Divya et al. [53] 2019	SVD + statistical features + wavelets	SDC	97.88%
			GMM	96.64%
8	Prabhakar et al. [54] 2020	Fuzzy-inspired statistical features	SVM–RBF (kernel) for CVD	95.05%
			RBF neural network-for normal	94.79%
9	Liu et al. [55] 2022	Time domain features	Deep convolutional neural network	85%
10	Ihsan et al. [56] 2022	HRV features and time domain features	Decision tree classifier	94.4%
11	Al Fahoum et al. [57] 2023	Time domain features	Naive Bayes	94.44% in first stage 89.37% in second stage
12	Rajaguru et al. [58] 2023	Statistical features	Linear regression	65.85%
13	As reported in this paper	Hilbert transform	Harmonic search classifier	98.31%

**Table 9 bioengineering-10-00678-t009:** Classifiers’ computational complexity among various dimensionality reduction techniques.

Classifiers	Optimization Techniques				
	Hilbert	NLR	ABC PSO	Cuckoo	Dragonfly
PCA	$O\left(m^{2}\right)$	$O(m^{2}\log m)$	$O\left(m^{5}\right)$	$O(2m^{2}\log m)$	$O(4m^{2}\log m)$
EM	$O\left(m^{2}\right)$	$O(m^{2}\log m)$	$O\left(m^{5}\right)$	$O(2m^{2}\log m)$	$O(4m^{2}\log m)$
Logistic regression	O(mlogm)	O(2mlogm)	$O(m^{4}\log m)$	O(4mlogm)	O(8mlogm)
GMM	$O\left(m^{2}\right)$	$O(m^{2}\log m)$	$O\left(m^{5}\right)$	$O(2m^{2}\log m)$	$O(4m^{2}\log m)$
Bayesian LDC	$O\left(m^{3}\right)$	$O(m^{3}\log m)$	$O\left(m^{6}\right)$	$O(2m^{3}\log m)$	$O(4m^{3}\log m)$
Firefly	O(mlogm)	$O(2m^{2}\log m)$	$O(m^{4}\log m)$	$O(m\log m^{2})$	O(8mlogm)
Harmonic search	$O\left(m^{3}\right)$	$O(m^{3}\log m)$	$O\left(m^{6}\right)$	$O(2m^{3}\log m)$	$O(4m^{3}\log m)$
DFA (weighted)	$O\left(m^{2}\right)$	$O(m^{2}\log m)$	$O\left(m^{5}\right)$	$O(2m^{2}\log m)$	$O(4m^{2}\log m)$
PAC Bayesian l earning	$O\left(m^{3}\right)$	$O(m^{3}\log m)$	$O\left(m^{6}\right)$	$O(2m^{3}\log m)$	$O(4m^{3}\log m)$
KNN-PAC Bayesian	$O\left(m^{4}\right)$	$O(m^{4}\log m)$	$O\left(m^{7}\right)$	$O(2m^{4}\log m)$	$O(4m^{4}\log m)$
SDC	$O\left(m^{2}\right)$	$O(m^{3}\log m)$	$O\left(m^{5}\right)$	$O(2m^{2}\log m)$	$O(4m^{2}\log m)$
Detrend SDC	$O\left(m^{3}\right)$	$O(m^{4}\log m)$	$O\left(m^{6}\right)$	$O(2m^{3}\log m)$	$O(4m^{3}\log m)$

## Data Availability

The data that support the findings of this study are available from the corresponding author upon reasonable request.

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
