# Peer review of "Machine Learning Techniques for the Performance Enhancement of Multiple Classifiers in the Detection of Cardiovascular Disease from PPG Signals"

_bioengineering, 2023, doi:10.3390/bioengineering10060678_

Round 1

Reviewer 1 Report

Comments to Authors (General)

·       The authors should cite the recent works which could be found in the link.

Tun, H. Photoplethysmography (PPG) Scheming System Based on Finite Impulse Response (FIR) Filter Design in Biomedical Applications. Int. J. Electr. Electron. Eng. Telecommun. 2021, 10, 272–282. Available online: http://www.ijeetc.com/index.php?m =content&c=index&a=show&catid=213&id=1525

·       The parameter of “b” in line 287 shall have to represent the specific name.

·       All equations shall have to be rechecked again.

·       All data in the tables and text shall have to be rechecked again.

·       The up to date summarization of the recent research works should be given.

Comments to Authors (Specific)

·       The authors shall have to revise the Figure 1. Organization of the CVD Detection from PPG Signals. There is no decision block in that diagram. The computerized system shall have to operate automatically according to the complete decision making process for the overall procedures.

·       The authors shall have to explain the threshold value H?(x) for the classifier under Logistic Regression (LR) is 0.5.

·       In line 731 and 732, “The lowest GDR of 40.755% is produced by the Firefly classifier under the ABC-PSO DR Method.” The value in the table 6 is 40.715. Which one is correct?

·       The authors shall have to rechecked in detail for section 5. Results and Discussion. There are many mistakes in that section.

·       The table.7 is very important and the authors made a lot of mistake to cite the recent works by comparing with the present work. The citation no in the table did not match in the text from line no 784 to 840. The authors shall have to modify carefully again.

·       The authors shall have to discuss the limitation of the works with logical expressions.

·       The future research work in conclusion shall have to be given the reason for using such kind of technology.

·       The authors shall have to mention the robustness of the proposed model.

Reviewer 2 Report

The authors present an interesting work.  The manuscript is well presented and organized, however, the following issues should be addressed:

a. I would suggest the authors to reduce the provided information about the classifiers and the metrics.  It would be interesting to provide information about the parameters values that were used and how they were determined.

b. the authors should check the references inside the paper, especially in section 5.3. Summary of Previous works for detection of CVD classes.  For examples they mention "Sunil Kumar et al [46] extracted the features from the PPG signal of a single patient who had a respiratory disorder, and the extracted features were classified with a GMM classifier and obtained an accuracy of 98.97%", however in Table 7 they mention Prabhakar,S.K. et al. [50].

c. a qualitative comparison between methods reported in rows 4 and 10 (proposed mehtod) of Table 7 is needed to highlight the contribution of the manuscript.

Reviewer 3 Report

The article seems interesting but has many shortcomings. The authors focused on a small dataset - 41 cases which may significantly affect the quality of the results. The authors use formulas in the article without criticism of their relevance, e.g. Table 1 contains obvious definitions. Equations 1-6 too, and in addition, their formatting needs corrections, e.g. the function sgn is defined using normal font and then it is used in italics. 

In general, instead of focusing on their own results, the authors expanded the article by describing the basics of the methods used. I believe that the atricle needs significant optimization to make it more readable and clearly present the results achieved.

Reviewer 4 Report

The paper is well-structured and comprehensively analyzes various dimensionality reduction techniques and classifiers for detecting Cardio Vascular Disease (CVD) from PPG signals. The authors have made a commendable effort to compare the performance of different techniques and identify the most suitable combination for the given problem.

Comments:

The authors have provided an extensive literature review in the introduction section, which helps readers understand the context of the research and its significance. However, it would be helpful if the authors could also provide a brief summary of the main contributions of this work, especially those that set it apart from previous studies.

It would be helpful if the authors could clarify the selection criteria for the five dimensionality reduction techniques and twelve classifiers used in this study. A brief explanation of why these specific techniques and classifiers were chosen would provide more insight into the research methodology.

The authors have detailed descriptions of the various dimensionality reduction techniques and classifiers used in the study. However, for clarity, it would be helpful to include brief definitions of each technique and classifier when they are first introduced in the text.

The paper's results section is comprehensive, providing tables and figures to help readers understand the performance of various techniques and classifiers. However, it would be helpful to include a brief interpretation of the results in the text. This would aid readers in understanding the key takeaways from the data and how the authors arrived at their conclusions.

The conclusion section provides a good summary of the paper's findings, but it could be improved by highlighting the study's practical implications. For example, the authors could discuss the potential impact of their findings on clinical practices and the development of PPG-based diagnostic tools for CVD.

The resolution of Figure 1 is low, which makes it difficult to understand the details of the organization of CVD detection from PPG signals. The authors should improve the resolution and readability of this figure to provide better visual support for the paper's methodology.

Recommendations:

1) Include a brief summary of the paper's main contributions in the introduction section.

2) Clarify the selection criteria for the dimensionality reduction techniques and classifiers used in the study.

3) Provide brief definitions of each technique and classifier when first introduced in the text, also defining the corresponding acronym (e.g., ABC-PSO).

4) Include a brief interpretation of the results in the results section to aid reader understanding.

5) Highlight the practical implications of the study in the conclusion section.

6) Improve the quality of Figure 1.

Round 2

Reviewer 3 Report

Regarding the explanation of the number of patients and the length of the signals, the information is heavily misleading. This information should also be included in the abstract, because the number of patients is irrelevant since we divide the signals into finer pieces according to the normal, not-normal classification. The number of signals used for analysis should be given.

From my point of view, unfortunately, little has been corrected to straighten out the perception of the article. There are countless editing errors in the equations presented. "for" is in italics in equation 6, "if" is in italics and is glued with n in equation 10.... My comments about the sgn function were taken in the opposite direction (function names in formulas are generally written in regular font and not in italics which is reserved by default for variables). And the worst thing is that from formula 39 variables and functions are already all written in regular font. I do not see the use of presenting a description of all the basic formulas while none of them are referenced in the text.  

I ask again: how do the formulas in Table 1 differ from the standard definitions of basic descriptive statistics?

In my opinion, the authors confused the concept of a review article with an original report and  that they did not prepare any of them reliably. They described all the methods used but, as I mentioned in terms of formulas, in a very chaotic way. For a person familiar with the subject, it is chaotic - as I pointed out the shortcomings earlier - a conglomeration of different information, for a person who wants to learn something a strongly unhelpful approach. Equation 15 is not explained at all what "~" is, but on the other hand the formula for the arithmetic mean is given in Table 1. 

To all these elements, the authors decided to include their results. The reasoning must be clear and the describing formulas consistent. 

Unfortunately, in my opinion the document is not prepared for publication.

Reviewer 4 Report

The authors have taken the time to carefully revise their manuscript and have made significant improvements based on the feedback provided by the reviewers. They have addressed all of the concerns the reviewers raised and provided clear and concise responses to their comments.

Author Response

Thank You for your comments.